# Donor and host photoreceptors engage in material transfer following transplantation of post-mitotic photoreceptor precursors

R.A. Pearson[1,*], A. Gonzalez-Cordero[1], E.L. West[1,**], J.R. Ribeiro[1,**], N. Aghaizu[1,**], D. Goh[1], R.D. Sampson[1], A. Georgiadis[1], P.V. Waldron[1], Y. Duran[1], A. Naeem[1], M. Kloc[1], E. Cristante[1], K. Kruczek[1], K. Warre-Cornish[1,†], J.C. Sowden[2], A.J. Smith[1] & R.R. Ali[1,3,*]

Photoreceptor replacement by transplantation is proposed as a treatment for blindness. Transplantation of healthy photoreceptor precursor cells into diseased murine eyes leads to the presence of functional photoreceptors within host retinae that express an array of donor-specific proteins. The resulting improvement in visual function was understood to be due to donor cells integrating within host retinae. Here, however, we show that while integration occurs the majority of donor-reporter-labelled cells in the host arises as a result of material transfer between donor and host photoreceptors. Material transfer does not involve permanent donor–host nuclear or cell–cell fusion, or the uptake of free protein or nucleic acid from the extracellular environment. Instead, RNA and/or protein are exchanged between donor and host cells in vivo. These data require a re-evaluation of the mechanisms underlying rescue by photoreceptor transplantation and raise the possibility of material transfer as a strategy for the treatment of retinal disorders.

[1] Department of Genetics, University College London Institute of Ophthalmology, 11-43 Bath Street, London EC1V 9EL, UK. [2] Stem Cells and Regenerative Medicine Section, 30 Guilford Street, London WC1N 1EH, UK. [3] NIHR Biomedical Research Centre at Moorfields Eye Hospital NHS Foundation Trust and UCL Institute of Ophthalmology, City Road, London EC1V 2PD, UK. * These authors jointly supervised this work. ** These authors contributed equally to this work. † Present address: Kings College London, Strand, London WC2R 2LS, UK. Correspondence and requests for materials should be addressed to R.A.P. (email: rachael.pearson@ucl.ac.uk) or to R.R.A. (email: r.ali@ucl.ac.uk).

Transplantation of healthy donor cells into diseased environments is a promising therapeutic strategy for a wide range of diseases. In the eye, despite different underlying causes, many degenerative disorders lead to the loss of the light-sensitive photoreceptors and blindness. At present, few clinically available treatments are capable of reversing this. Clinical trials for gene-supplementation therapy have shown promise for patients with known genetic defects[1–6], but photoreceptor replacement by transplantation is proposed as a broad treatment strategy applicable to many forms of retinal degeneration[7,8]. Photoreceptor replacement could be useful during the degenerative process, when some host photoreceptors remain, for diseases with an unknown aetiology. Alternatively, cell therapy may be applied as a treatment for end-stage disease, when little, if any, of the outer nuclear layer (ONL) of the host retina remains.

We, and others, have shown previously that the transplantation of stage-specific post-mitotic rod photoreceptor precursors carrying a transgenic green fluorescent protein (GFP) label results in the presence of GFP-positive (GFP$^+$) rod-like cells within the ONL where photoreceptors normally reside, in wild-type and degenerating murine host retinae[9–16]. These cells bear the morphological characteristics of mature photoreceptors, including synapse-like structures and outer segments. Importantly, when transplanted into different models of retinal degeneration, GFP$^+$ cells within the host ONL demonstrated robust levels of those proteins that were genetically absent in the host photoreceptors[9–13]. Moreover, single-cell[12] and whole-retinal[9,17,18] recordings showed these GFP$^+$ cells to be light-responsive in a manner very similar to that of normal wild-type photoreceptors. The presence of these cells also correlated with visually evoked activity in the visual cortex and behaviour, when present in sufficiently large numbers[12]. Strikingly, the developmental stage of the donor cell at the time of transplantation is important; transplantation of embryonic or adult photoreceptors led to poor integration of cells within the host ONL[9,16], a finding that holds true for both postnatally derived donor cells[9] and photoreceptor precursors derived from three-dimensional (3D) differentiation of embryonic stem (ES) cells[14,19]. Altogether, these data provided evidence of robust rescue of photoreceptor function following the transplantation of healthy photoreceptors.

Functional rescue of different tissues in a variety of disease models by stem cells has been widely reported. However, what was previously thought to be stem cell integration and differentiation has since been shown to be cell–cell fusion between the donor stem cell and the host cell in some studies[20]. Classic cell fusion involves the combining of two cells and their nuclei. The two nuclei may remain as separate entities[21,22] or may slowly fuse to form a single nucleus. The process of nuclear fusion between mature cells is typically slow and takes many days to weeks[23,24]. In some cases, a third cell is involved; for example, transplantation of neural stem cells into the rodent cortex results in their fusion with mature pyramidal neurons, in a two-step process that appears to be mediated by microglia[25]. Such reports are not uncommon where the donor cell is a stem cell. There are very few examples, however, where fusion occurs between donor and host neurons when both are post-mitotic[20].

Previously, we, and others, addressed the possibility of cell fusion after transplantation of post-mitotic photoreceptors into the adult host retina using two techniques. In the first, a permanent nuclear label, BrdU, was introduced into a proportion of donor cells before transplantation[9]. We found examples of GFP$^+$ photoreceptors within the host retina that bore a single BrdU$^+$ nucleus, providing strong evidence that these were donor cells that had migrated into the host retina.

Other apparently integrated cells were BrdU$^-$ but as only a small proportion of the starting donor cell population carried the label, this was not unexpected. In the second, GFP$^+$ donors were transplanted into CFP(cyan fluorescent protein)-transgenic recipient mice. Confocal imaging found GFP$^+$ cells within the host retina with GFP$^+$/CFP$^-$ inner segments[9]. Ader and colleagues[10] reported similar results by using a viral vector to label their donor population with tdTomato and transplanting into eGFP-transgenic recipients. Others have heralded this dual-fluorescent donor/recipient technique as a gold standard for assessing cell fusion[26]. Moreover, analysis of transplanted retinae at different stages post transplantation, ranging from 48 h to 6 weeks, revealed the presence of stereotyped morphologies of donor cells approaching and extending processes to the host retina, before moving across the outer limiting membrane (OLM)[27], a physical barrier whose disruption can lead to significantly increased numbers of donor-reporter-labelled cells within the host retina[13,28,29]. Altogether, these findings have been regarded by the ocular field as strong evidence supporting the hypothesis that donor cells migrate into the recipient retina and mature and differentiate in situ.

As we have continued to research photoreceptor transplantation in increasing detail, we have made observations that suggest that donor cell migration and integration is not the only mechanism involved. Below, we present data using a wide variety of techniques that indicate that post-mitotic donor and host photoreceptors can engage in the transfer of either RNA and/or protein, resulting in the robust presence of a wide variety of donor-specific proteins in host cells. While the precise transfer mechanism remains unclear, it does not involve classic cell–cell nuclear fusion.

## Results

**Real-time imaging of integration events**. FAC-sorted GFP$^+$ photoreceptor precursor cells from postnatal day 8 (P8) *NrlGFP B6.Cg-Tg(Nrl-EGFP)1Asw/J* (*NrlGFP*) mice[30] were transplanted into adult *Prph2$^{rd2/rd2}$* (ref. 31) hosts. Explanted host retinae were labelled with Mitotracker Orange CMTMRos to visualize the host retinal structure[32,33] and associated donor cell mass and imaged 72 h post transplantation using 2-photon real-time imaging. Some donor cells appear to move into the host retinae over a period of several hours (Fig. 1; Supplementary Movie 1). Typically, donor cells initially locate to the interphotoreceptor matrix and appear to extend a process toward the OLM, before moving into the host ONL. Movement into the host retina was restricted to the first 1–2 photoreceptor rows and deeper penetration was not observed, although it is possible that such migration occurs over a longer time period than was possible to image here. These data support the occurrence of donor cell migration into the host retina, very similar to that reported for fixed tissue time series[27].

**Exchange of reporters between donor and host photoreceptors**. In a complementary series of experiments aiming to evaluate donor–host cell interactions, we repeated the fluorescent reporter transplants that we, and others, reported previously[9,10], but this time using two different fluorescent labels and analysis by confocal microscopy and flow cytometry. *NrlGFP* donors were transplanted into adult *B6.Cg-Tg(CAG-DsRed*MST)1Nagy/J* (*DsRed*) transgenic hosts, which ubiquitously express the fluorescent reporter, DsRed. At 5–6 weeks post transplantation, retinal sections were imaged using confocal microscopy and assessed for localization of GFP$^+$ cells within the *DsRed* host ONL (Fig. 2). Of 157 GFP$^+$ cells (*N* = 5 retinae) examined, 36% of inner segments were identified by two assessors as showing an increase

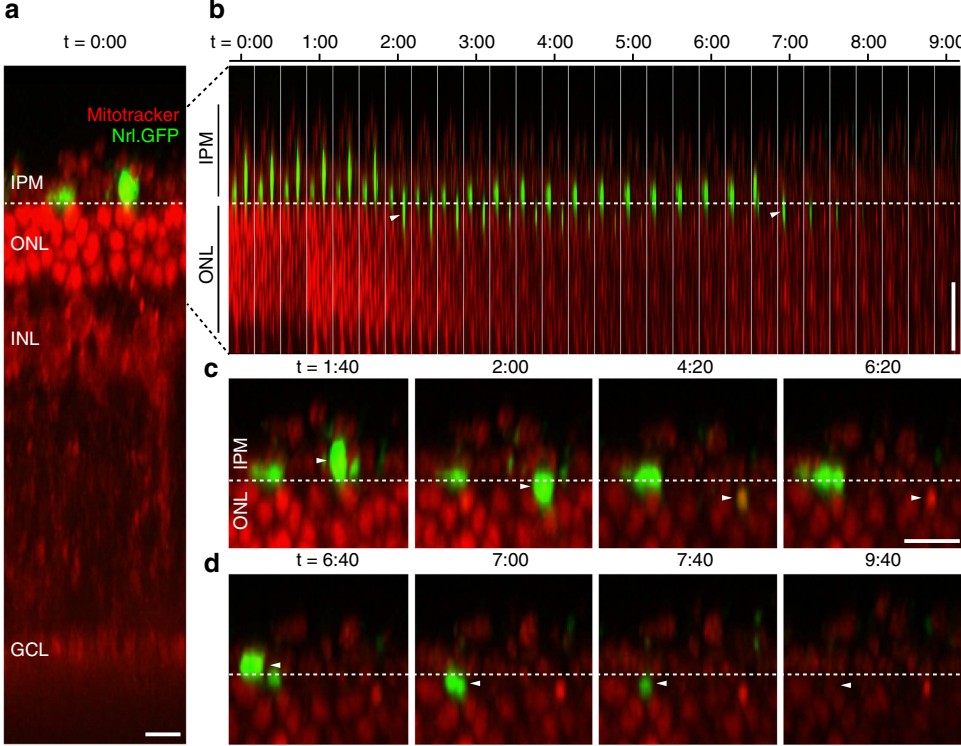

**Figure 1 | Real-time imaging of transplanted donor precursor cells migrating into host retinae.** *NrlGFP* post-mitotic photoreceptor precursor donor cells (*green*) were transplanted into 12 weeks old *Prph2^{rd2/rd2}* mice and explanted retinae were examined by real-time 2-photon fluorescence live imaging 3 days post transplantation (retinae, including donor and host cells, were acutely labelled with Mitotracker Orange CMTMRos (*red*) before the recording). (**a**), At the start of the time-lapse recordings, sub-retinally transplanted rod precursor cells were frequently observed at the level of the IPM or apical to the retinal tissue. A dashed line (*white*) was inserted at the presumptive level of the OLM. (**b**), Time course of rod precursor cell movement from the IPM into the ONL; penetration through the OLM (*white dashed line*) is indicated by *white arrowheads* (see also Supplementary Movie 1). (**c,d**), High magnification views of time frames shown in (**b**) depicting migration of the right (**c**) and left (**d**) rod precursor cell into the ONL at selected time points. Note that while GFP fluorescence gradually reduced over the imaging period, Mitotracker Orange CMTMRos labelling, which is present in both donor and host cells, persisted in its absence. Scale bars, 10 μm.

in GFP fluorescence and a concomitant decrease in DsRed fluorescence (GFP$^+$/DsRed$^-$; Fig. 2a,d), indicative of integrated cells. However, in 60% of cells the DsRed signal was either unchanged or increased relative to neighbouring cells (GFP$^+$/DsRed$^+$; Fig. 2a,e). Although stringent confocal settings were used (see the 'Methods' section), the dense packing of rod photoreceptors means that there is very little cytoplasm, making assessments of co-localization of cytoplasmic labels prone to error, particularly overestimation of co-localization[34]. Nevertheless, when examining cell bodies of the same population, 21% were GFP$^+$/DsRed$^-$ (Fig. 2b,f), leaving 76% of GFP$^+$ cells being additionally DsRed$^+$ (Fig. 2b,g). The remaining cells ($<4\%$) represent those for which there was no consensus between assessors. Interestingly, when the GFP$^+$/DsRed$^-$ cells (integrated) were assessed with respect to their position in the host ONL, there was significant trend for them to be located within the first two photoreceptor rows, closest to the OLM (Fig. 2h,i). In addition, we examined those donor cells that failed to integrate but remained in the subretinal space (SRS). Most expressed GFP alone but a small number also expressed DsRed, albeit at low levels (Fig. 2j). These observations suggest that while some donor photoreceptors integrate into the host ONL, donor-reporter RNA and/or proteins might also be able to move not only from donor to host photoreceptors but also vice versa.

The difference in the proportion of GFP$^+$/DsRed$^+$ cell bodies, compared with inner segments, within the same population of cells demonstrates the limitations of this method of analysis. We repeated the experiment, this time dissociating

the isolated host neural retinae and assessing the resulting cell populations using flow cytometry (Fig. 3 and Supplementary Fig. 1). The gates were set on the basis of *wildtype*, *DsRed* and *NrlGFP* controls (Fig. 3b–d). Of 18 host retinae examined, the total number of GFP$^+$ cells collected per host eye ranged between 120 and 10,575 cells (mean $= 2,130 \pm 2,772$ cells; Fig. 3a). Of these, 18.7% ($\pm 24.9$; median value $= 4.7\%$) were GFP$^+$/DsRed$^-$, however 81.4% ($\pm 24.8$; median value $= 95.3\%$) of GFP$^+$ cells were also DsRed$^+$. GFP$^+$/DsRed$^-$ cells had slightly higher levels of GFP when compared with GFP$^+$/DsRed$^+$ cells, as demonstrated by mean fluorescence intensity (Fig. 3e,f and Supplementary Fig. 1). Taken together with the confocal data, the GFP$^+$/DsRed$^-$ population likely corresponds to integrated cells, although a small proportion may reflect donor cells located in the SRS that had adhered to the neural retina. We excluded the possibility that GFP$^+$/DsRed$^+$ cells included resident or infiltrating macrophages that had phagocytosed GFP, by using CD45 staining. Less than 0.016% of GFP$^+$/DsRed$^+$ cells co-stained with CD45 in any given sample ($N = 8$ retinal samples; Fig. 3g), confirming that they were not macrophages and/or other white blood cells.

These analyses confirm the presence of GFP$^+$/DsRed$^-$ cells within the host ONL that are most likely donor cells that migrate into the recipient retina. However, the high proportion of GFP$^+$/DsRed$^+$ cells in both experiments suggests a second mechanism, herein termed material transfer, which can also result in the labelling of host photoreceptors with a genetic marker derived from the donor cells.

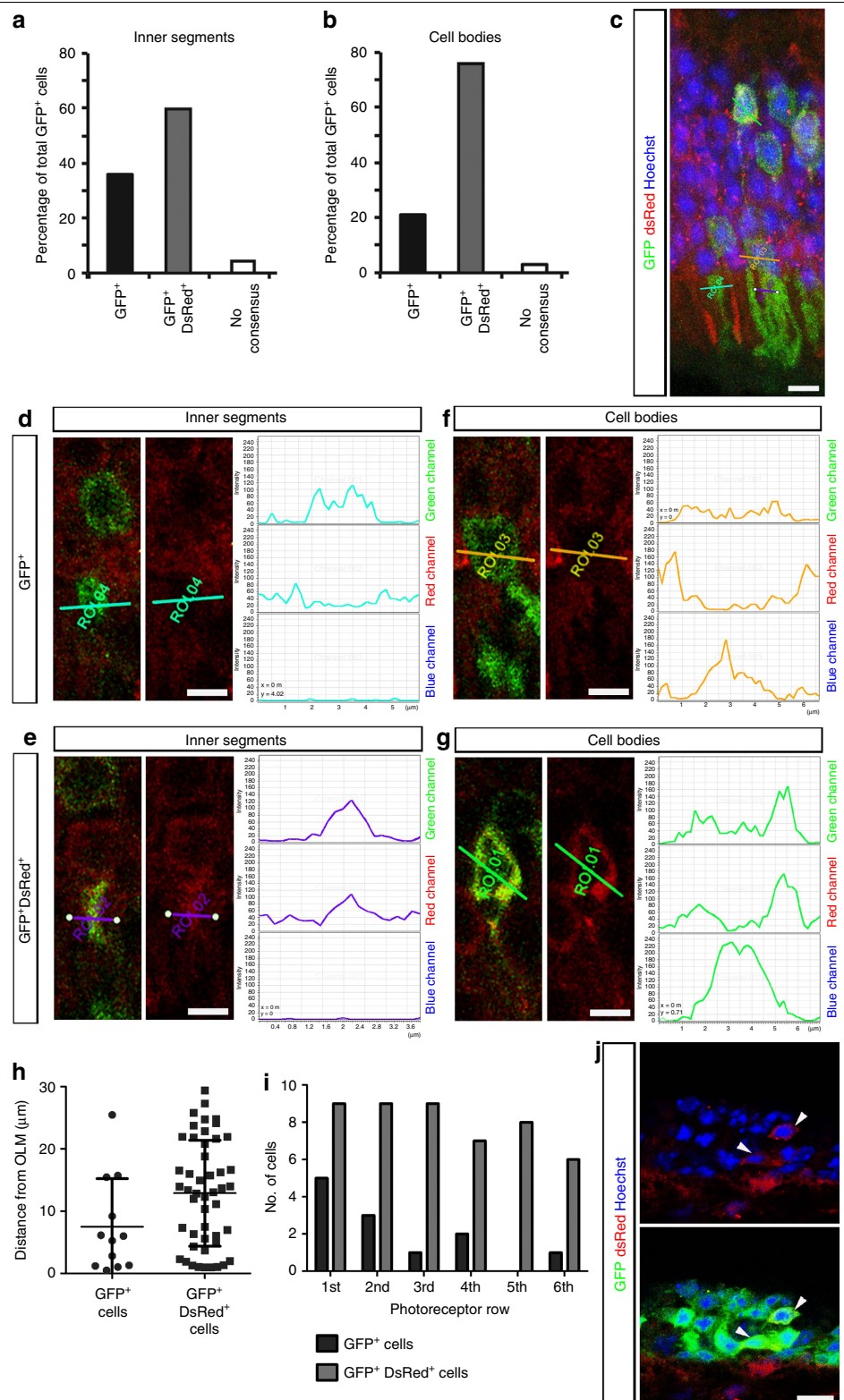

**Figure 2 | Host and donor photoreceptor exchange fluorescent reporter proteins as assessed by confocal imaging.** *NrlGFP* post-mitotic photoreceptor precursor donor cells (*green*) were transplanted into *DsRed* (*red*) hosts and examined by confocal microscopy 5-6 weeks post transplantation. Nuclei are labelled with Hoechst (*blue*). (**a**, **b**), histograms showing quantification of the proportion of cells analysed (*n* = 157 cells; *N* = 5 retinae) that expressed GFP alone (GFP+) or GFP and DsRed (GFP+/DsRed+) when measuring inner segments (**a**) or cell bodies (**b**). (**c**), confocal projection image of a representative host retina. Lines represent regions of interest (ROIs) through cells and inner segments shown in (**d**–**g**). (**d**), (**e**) single confocal sections and the respective line plots through two inner segments and (**f**,**g**), two cell bodies shown in (**c**). (**h**,**i**), scatter plot (mean ± s.d.) and histogram showing the position of GFP+ and GFP+/DsRed+ photoreceptors with respect to the outer limiting membrane (OLM) of host retina. The relative positions of the two populations were significantly different (*P* = 0.0486 two tailed t-test with Welch's correction; *P* < 0.001 2-way ANOVA). (**j**), confocal projection image showing donor NrlGFP+ cells in SRS of DsRed+ host. *Arrows*, donor GFP+ cells that are DsRed+. Scale bars 5 μm (**c**), 3 μm (**d**) and 10 μm (**g**).

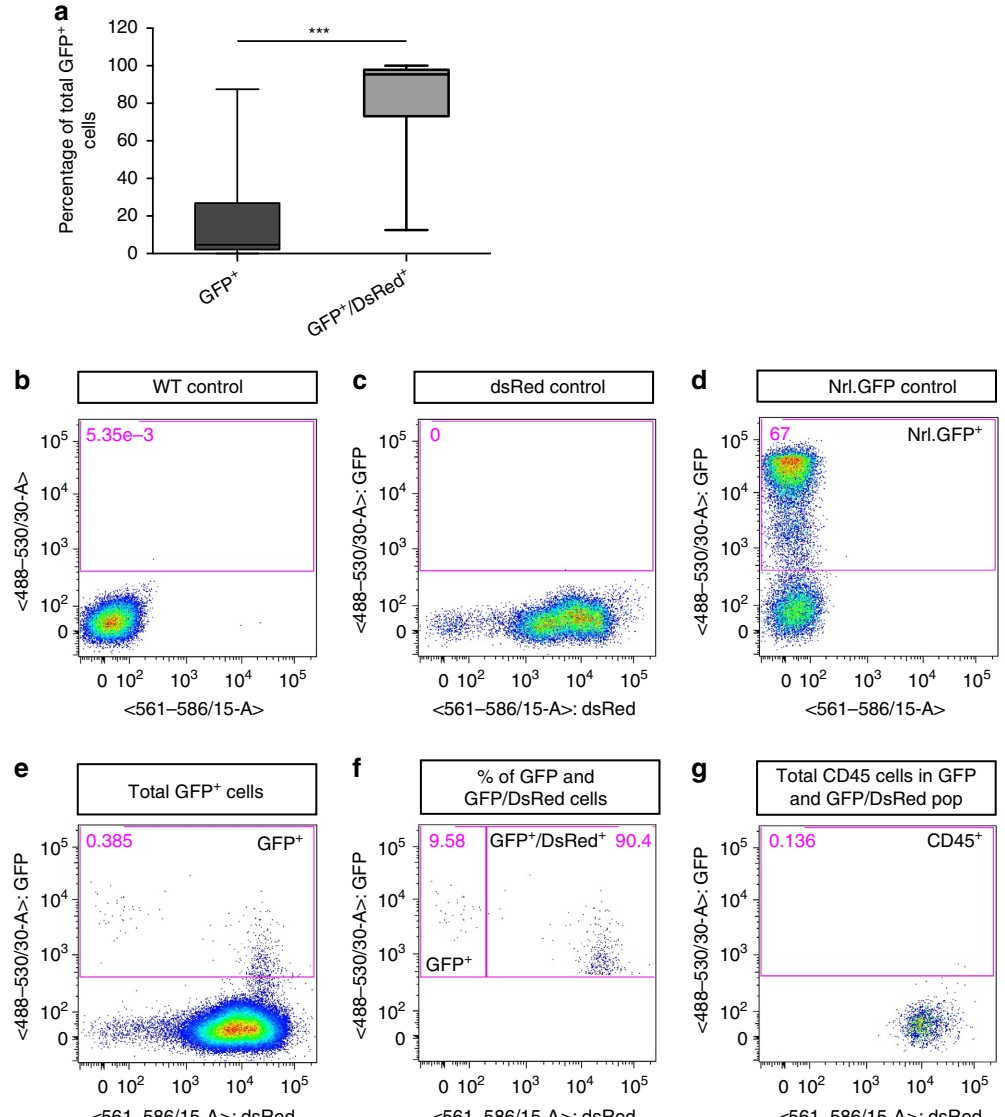

**Figure 3 | Host and donor photoreceptors exchange fluorescent reporter proteins as determined by flow cytometry.** *NrlGFP* post-mitotic photoreceptor precursor donor cells were transplanted into *DsRed* hosts and examined by flow cytometry 5-6 weeks post transplantation. (**a**), box (25–75% percentile) and whiskers ($_{min}$/max) plot showing median (line) % of GFP$^+$ only and GFP$^+$/DsRed$^+$ photoreceptors within each host retina ($N = 18$ retinae). ***$P < 0.001$, paired *t*-test. (**b–d**), representative flow cytometry plots for adult (**b**) wild-type (negative control), (**c**) *DsRed* (positive control) and (**d**) *NrlGFP* (positive control) retinae. *Pink box* shows gating for GFP$^+$ cells. (**e,f**), representative plots from an example of a host retina showing (**e**) % of total retinal cells that were GFP$^+$ (*pink box*) and (**f**) the proportion of these that were GFP$^+$ only (*left pink box*) or GFP$^+$/DsRed$^+$ (*right pink box*). (**g**), plot showing the proportion of CD45+ cells within the GFP$^+$ population shown in (**e,f**).

**Material transfer does not involve nuclear fusion**. Transfer of genetic information is typically associated with cell fusion, either with or without nuclear fusion. Previously, we, and others, reported that integrated cells only ever contain a single nucleus[9,10] and do so from the earliest stages examined post transplantation (48 h) (ref. 27). Here, we sought to re-examine the possibility of nuclear fusion for those host cells expressing donor-derived reporters. We examined >1,000 GFP$^+$ photoreceptors located within the ONL of transplanted *wild-type* or *Gnat1$^{-/-}$* host retinae ($N = 6$) stained for the nuclear envelope protein, LaminB[35]. Without exception, all GFP$^+$ cells had a single nuclear envelope and no evidence of polyploidy (Fig. 4a–c). The vast majority had highly condensed nuclei, typical of rod photoreceptors (>98%) (ref. 36), although cone-like profiles with multi-chromocentred single nuclei were seen occasionally.

A retrospective examination of previously published data[27] confirmed that integrated GFP$^+$ cells only ever contained a single nucleus from as early as 48 h post transplantation ($n = 159$ cells; $N = 5$ at 48 h post transplantation), arguing strongly against nuclear fusion, an event rarely seen between post-mitotic neurons[23]. To rule this out conclusively, we transplanted male *NrlGFP$^+$* cells into female wild-type hosts and performed Fluorescent *In Situ* Hybridization (FISH) against the Y-chromosome, at 5–6 weeks post transplantation (Fig. 4d–g). Y-chromosome probe staining was detected in 83 ($\pm 7$)% of photoreceptors in male *NrlGFP* eyes (positive control; Fig. 4d; $N = 4$), compared with 0 ($\pm 0$)% of cells in female *NrlGFP* eyes (negative control; Fig. 4e; $N = 3$). Positive labelling for the Y-chromosome was detected in 84 ($\pm 10$)% of donor cells within the sub-retinally located cell mass in transplanted eyes ($N = 8$).

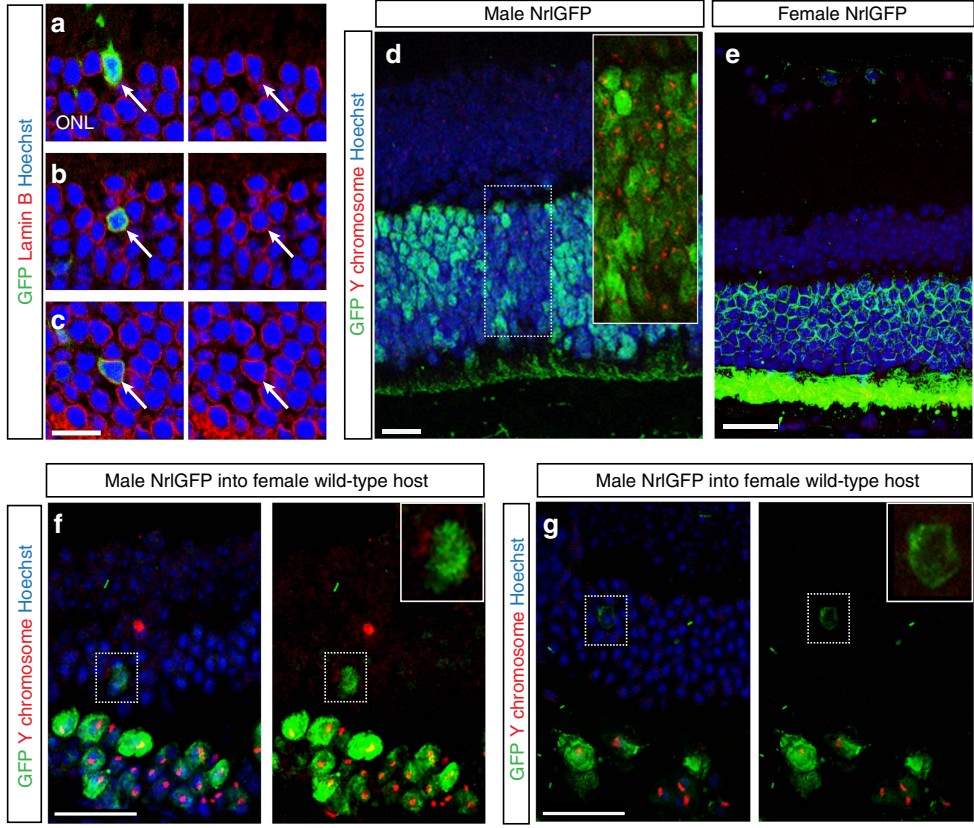

**Figure 4 | Material transfer is not the result of nuclear fusion. (a–c)**, representative confocal images of retinal sections from $Gnat1^{-/-}$ and wild-type retinae following transplantation of *NrlGFP* post-mitotic photoreceptor precursors (*green*), showing expression of Lamin B (*red*). GFP$^+$ cells (*arrows*) within the host ONL only ever presented a single nuclear envelope (*red*) and nucleus (labelled with Hoechst; *blue*). Right hand panels show nuclei and Lamin B staining only. Scale bars 10 μm (**d–g**), FISH for the male Y-chromosome. (**d**), Male and (**e**), female *NrlGFP* (*green*) retinal sections stained for the Y-chromosome (*red*), which appears as a red dot. *Insert*, Male GFP$^+$ cells with Y-chromosome staining. No Y-chromosome staining was seen in the female sections (**e**). (**f,g**), examples of GFP$^+$ cells within female host retinae 5-6 weeks post-transplantation of male *NrlGFP* donor cells, showing a GFP$^+$ cell that was positive for Y-chromosome staining (**f**, *insert*) and another that was GFP$^+$/Y-chromosome$^-$ (**g**, *insert*). Right hand panels show GFP and Y-chromosome staining only. Scale bars 25 μm

We also detected rare examples of Y-chromosome labelling in GFP$^+$ cells located within the recipient ONL (Fig. 4f). However, Y-chromosome staining was absent from the large majority of GFP$^+$ cells located within the ONL (Fig. 4g and Supplementary Fig. 2).

The presence of Y-chromosome$^+$ cells within the host retina, albeit in low numbers, together with previous observations in which donor cells with pre-labelled nuclei were present within host retinae[9] supports the occurrence of true donor cell integration. However, the marked difference between the number of GFP$^+$/Y-chromosome$^+$ donor cells within the SRS and the number of GFP$^+$/Y-chromosome$^-$ within the host ONL suggests that a significant proportion of GFP$^+$ cells within the host retina arise by material transfer via a mechanism that does not involve transfer of the donor cell nucleus.

**Donor and host photoreceptors exchange array of gene products.** The data above suggest that host photoreceptors can acquire reporter proteins that are genetically present only in the donor cell. We next considered whether this was a phenomenon particular to the fluorescent reporters used, or represents a more widespread mechanism. It has previously been shown that, in addition to the donor-derived fluorescent reporters, host cells display a variety of proteins that are otherwise missing from photoreceptors cells within the host ONL: for example, the photopigment rhodopsin is present in GFP$^+$ cells within the

ONL of $Rho^{-/-}$ mice[9,11,13,19], the structural protein Peripherin is found in GFP$^+$ cells in the $Prph2^{rd2/rd2}$ ONL[9,13,19] and rod α-transducin (encoded by the $Gnat1$ gene) is present in GFP$^+$ cells in the $Gnat1^{-/-}$ host retina[12], each one in its correct location and for many weeks post transplantation[27]. Indeed, the presence of GFP$^+$ cells in the host ONL has been observed as late as 1 year post transplantation[37].

We sought to determine how robust the apparent material transfer between donor and recipient cells is. We examined $Gnat1^{-/-}$ recipient mice that had received either only $NrlGFP^+$ donor cells or a mix of $NrlGFP^+$ donor cells and $DsRed^+$ donor cells (Fig. 5). By 6 weeks post transplantation, rod α-transducin, the protein absent from host photoreceptors, was found in >83% of GFP$^+$ cells located within the recipient $Gnat1^{-/-}$ ONL (Fig. 5a–c,e; $n = 138$; $N = 4$). These included isolated cells lying some significant distance away from the donor cells in the SRS (Fig. 5b). Importantly, this held true for both GFP$^+$ and DsRed$^+$ cells (Fig. 5f). In these experiments, based on the data presented above, we assume that a significant proportion of the GFP$^+$ cells located within the ONL are host, rather than donor, cells. Occasionally, host ONL cells were rod α-transducin$^+$ but GFP$^-$ (Fig. 5d). Altogether with previously published data[9,11,13,19], this demonstrates that the mechanism leading to material transfer between donor and host cells is sufficiently robust to result in the presence of a wide variety of donor-derived proteins, including many that are genetically missing from diseased host photoreceptors.

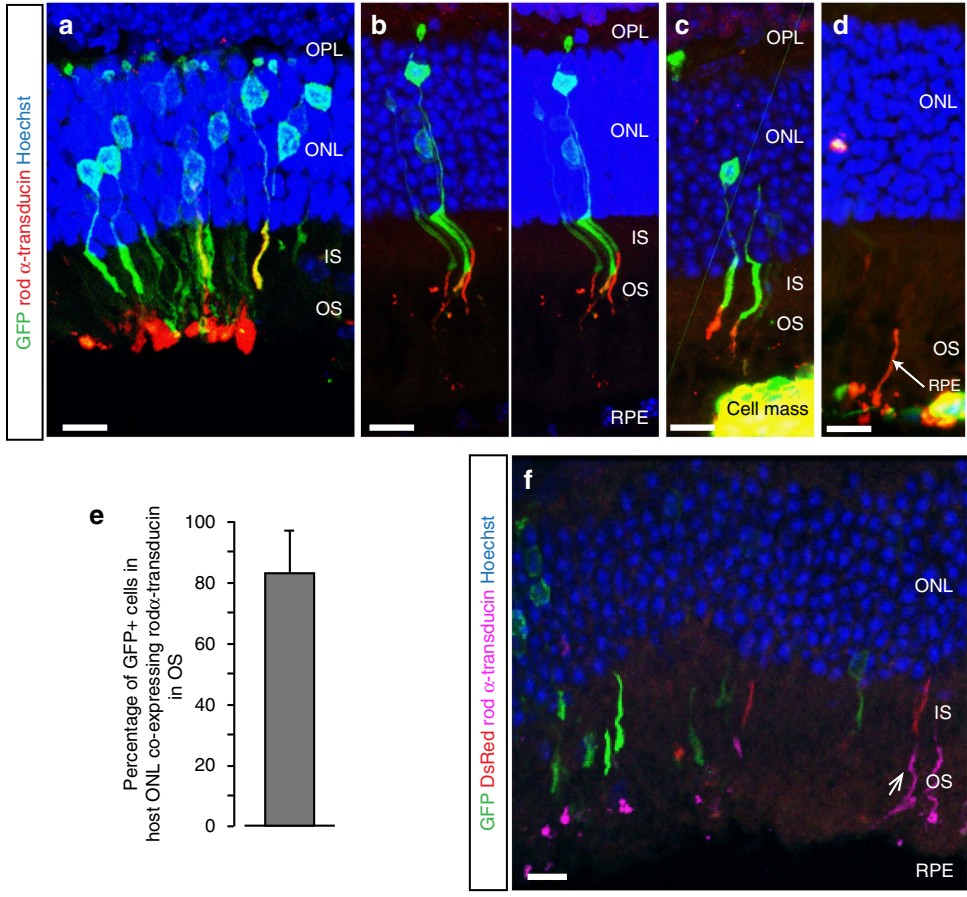

**Figure 5 | Material transfer permits robust exchange of an array of gene products including those genetically absent from host photoreceptors.** *NrlGFP* post-mitotic photoreceptor precursor donor cells were transplanted into adult *Gnat1*$^{-/-}$ hosts and immunostained for rod α-transducin 5-6 weeks post-transplantation (**a–c**). Confocal projection images of GFP$^+$ cells (*green*) in the ONL of host retinae stained for rod α-transducin (*red*). Nuclei were stained with Hoechst 33342 (*blue*). Note that in (**b**) the right hand panel shows the same image but with increased gain for the blue channel to show the nuclei of the RPE: No donor cells were present in the SRS overlying the GFP$^+$/rod α-transducin$^+$ cells. (**d**), Examples of GFP$^-$/rod α-transducin$^+$ host cells (*arrow*) were also seen. (**e**), Histogram showing the mean percentage ( ± s.d.) of GFP$^+$ photoreceptors within the host ONL that were rod α-transducin$^+$ ($n = 138$ cells; N = 4 retinae). **f**, rod α-transducin staining was seen in the outer segments of both GFP- and DsRed-reporter-labelled photoreceptors (*arrow*). Scale bars 10 μm.

**Material transfer does not involve uptake of free proteins**. We considered the possibility that free protein could be released into the environment by the transplanted donor cells that remain in the SRS and/or by resident macrophages and taken up by the host photoreceptors. Recombinant eGFP (rEGFP) was injected into the SRS of *Gnat1*$^{-/-}$ recipient mice and eyes were examined at 48 h, 1, 2 and 6 weeks post-injection (Fig. 6a–f). Robust GFP fluorescence was seen throughout the SRS and in the segment region at 48 h post-injection. GFP was reduced at 1 week post-injection, but still widespread throughout the SRS, and largely absent from 2 weeks onwards (Fig. 6a–d). This time course corresponds well with the reported half-life for eGFP[38]. Despite effectively flooding the retina with rEGFP, we observed only very few weakly GFP$^+$ cells within the host ONL ($12 \pm 9$ cells for stained sections, Fig. 6a,e; $0 \pm 0$ in unstained serial sections; $N = 6$; Fig. 6a–f) at 48 h post-injection. By contrast, no GFP cells were seen in the recipient ONL at 1 ($N = 5$), 2 ($N = 8$) or 6 ($N = 4$) weeks post-injection (Fig. 6b–f). These results indicate that host cell labelling does not readily result from the uptake of free protein from the extracellular environment.

GFP has a half-life of ~26 h (ref. 39), while donor cells can remain in the SRS for many months[37]. To determine the specificity of material transfer seen in the presence of photoreceptor precursors, we next examined the effect of transplanting other populations of GFP$^+$ cells into the SRS. Transplantation of GFP$^+$ fibroblasts induced macrophage infiltration and rejection of the transplanted cells in all eyes examined at 2 ($N = 4$) and 6 ($N = 8$) weeks post transplantation. Although autofluorescent macrophages were present, on no occasion was GFP signal found within the recipient ONL (Fig. 6g,h).

We have previously reported that transplanted GFP$^+$ retinal progenitor cells (RPCs) survive in the SRS but do not integrate into the host retina[9]. We repeated this experiment, this time transplanting E11.5 *cba.GFP*$^{+/-}$ RPCs, prepared in the same way as postnatally derived donors, into adult *Gnat1*$^{-/-}$ recipients. Donor cells survived in the SRS, but in 3 out of 4 eyes examined, no GFP$^+$ cells were found in the recipient ONL and just 3 were found in the ONL of the fourth eye (0.75 cells $\pm 1.5$; $N = 4$) at 6 weeks post transplantation (Fig. 6i,j).

Altogether, these data indicate that material transfer to photoreceptors in the host ONL is a specific property of stage-specific donor photoreceptor precursors (and not other cell types) and does not result from uptake of free-floating protein.

**Material transfer involves donor and host cell interaction**. The use of rEGFP suggests that material transfer is not the result of uptake of free-floating GFP protein by host photoreceptors.

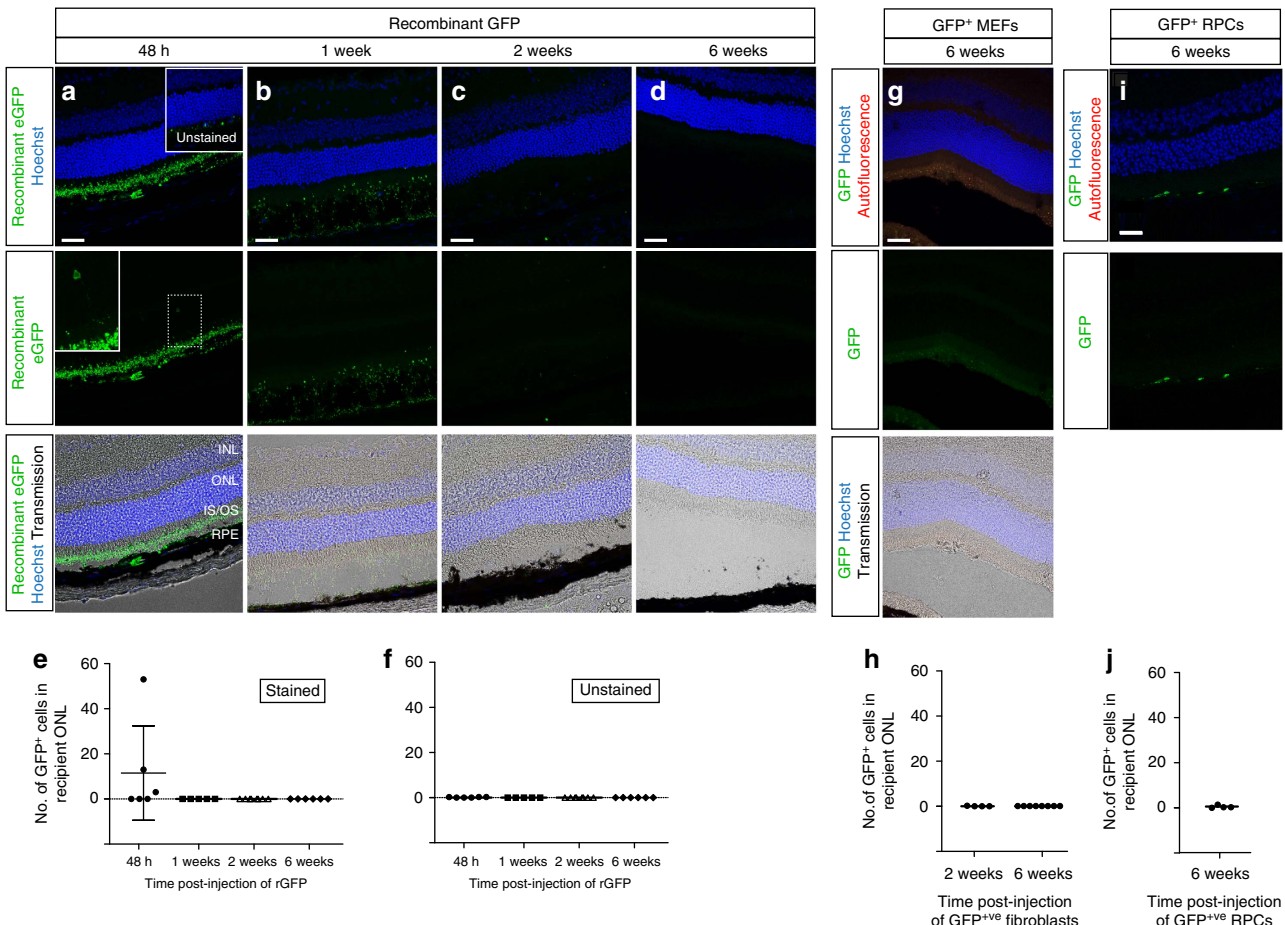

**Figure 6 | Material transfer is specific to photoreceptor precursors and does not result from uptake of free proteins from environment.** (**a**–**d**), confocal projection images of adult retinal sections taken at 48 h, 1 week, 2weeks and 6 weeks post-injection of rEGFP and stained with anti-GFP antibody (*green*). *Insert*, Very few GFP$^+$ cells were seen; those that were had normal rod photoreceptor morphology. (**e**,**f**) Scatter plots showing the mean ( ± s.d.) number of GFP$^+$ cells within the host ONL at each time point in stained and unstained serial sections. (**g**), confocal projection images of retinal sections taken at 6 weeks post-injection of GFP$^+$ fibroblasts. (**h**), scatter plot showing the number of GFP$^+$ cells within the host ONL. (**i**), confocal projection images of retinal section taken at 6 weeks post-injection of E11.5 GFP$^+$ retinal progenitor cells (RPCs). (**j**), scatter plot showing the number of GFP$^+$ cells within the host ONL. For panels (**a**)-(**d**) and (**g**),(**h**), images from top to bottom show individual and/or combined channels for the same region of interest. Scale bars 50 μm.

We next compared the distribution of labelled cells following transplantation of a mixed population of donor photoreceptor precursors taken from both *NrlGFP* and *DsRed* mice. Post transplantation, GFP$^+$ and DsRed$^+$ cells were randomly distributed within the host ONL (Fig. 7a,b). Strikingly, we observed examples where both GFP and DsRed were localized in a single photoreceptor within the host ONL (Fig. 7c). These were rare in number (1.9% of all cells examined; $n = 500$, $N = 10$), but a similar pattern was also observed in donor cells that remained within the SRS. The majority of the donor cells in the SRS were either GFP$^+$ or DsRed$^+$, but 6.0% of cells ($n = 300$, $N = 10$) colocalised both fluorescent signals (Fig. 7d,e). In all cases, labelled cells bore a single nucleus. The presence of dual-labelled cells within the host retina might be explained either by material transfer leading to the exchange of GFP and DsRed RNA, and/or protein between donor cells before one cell subsequently integrating, or by RNA and/or protein from both *DsRed* donors and *NrlGFP* donors being passed to a host photoreceptor cell.

The presence of even rare examples of dual-labelled cells is surprising. The observation that the majority of labelled cells were discrete entities, labelled with GFP or DsRed and located in the ONL, combined with the absence of fluorescence in the extracellular matrix, suggests that material transfer occurs specifically between donor and host photoreceptors. This could be via a direct physical cell–cell interaction or indirectly, by extracellular trafficking of (presumably packaged) RNA and/or protein between cells. The relatively low number of reporter-labelled host photoreceptors, compared with the total number of host photoreceptors, indicates either that relatively few host cells can undergo material transfer or that there is limited/transient supply of that material.

**Material transfer does not require sustained cell–cell fusion.** Recent studies have indicated that neural stem cells, among others, can fuse with adult neurons in a process that requires microglia[25]. In these studies, careful examination of the adult neuron always revealed the presence of a second, reporter-positive cell adjacent to it. It is possible that material transfer similarly results from the direct physical interaction between donor and host photoreceptors. Examination of retinal sections from transplanted eyes shows that many GFP$^+$ cells within the host ONL lie immediately below the donor cell mass (Fig. 8a). Moreover, their outer segments, by default, often terminate at its edge suggesting a potential physical interaction. However, numerous examples could be found of GFP$^+$ cells located in

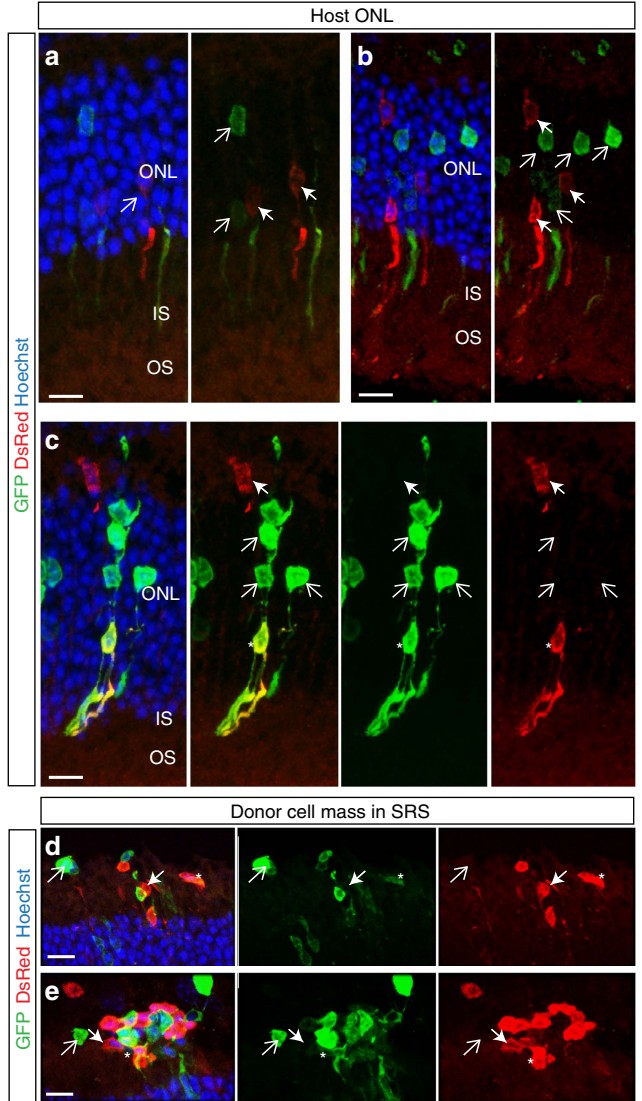

**Figure 7 | Material transfer involves interaction between donor and host photoreceptors** NrlGFP$^{+}$/CD73$^{+}$ (green) and DsRed$^{+}$/CD73$^{+}$ (red) donor cells were transplanted into adult wild-type and Gnat1$^{-/-}$ recipients. (**a–c**), confocal projection images showing heterogeneous mixes of DsRed$^{+}$ (closed arrow) and GFP$^{+}$ (open arrow) cells within the host ONL. Occasionally, a photoreceptor within the host ONL bore both GFP and DsRed labels (asterisk). Nuclei are labelled with Hoechst (blue). (**d,e**), confocal projection images showing a similar finding for donor cells in the SRS. The majority expressed either GFP (open arrow) or DsRed (closed arrow), but some expressed both reporters (asterisk). For all panels, images from left to right show individual and/or combined channels for the same region of interest. Scale bars 10 µm.

the host ONL without any GFP$^{+}$ donor cells within the overlying SRS (Fig. 8b–d). When examined using 3D reconstruction, these cells had near-normal photoreceptor morphology with no evidence of physical attachment to a second cell (Fig. 8c,d). These data do not rule out the occurrence of transient physical interactions, but do indicate that a sustained physical connection between donor and host photoreceptor is not essential for material transfer.

**Material transfer is widespread transient and repeated**. The Cre/LoxP system has been used in a number of transplantation paradigms to investigate the occurrence of cell fusion. Here,

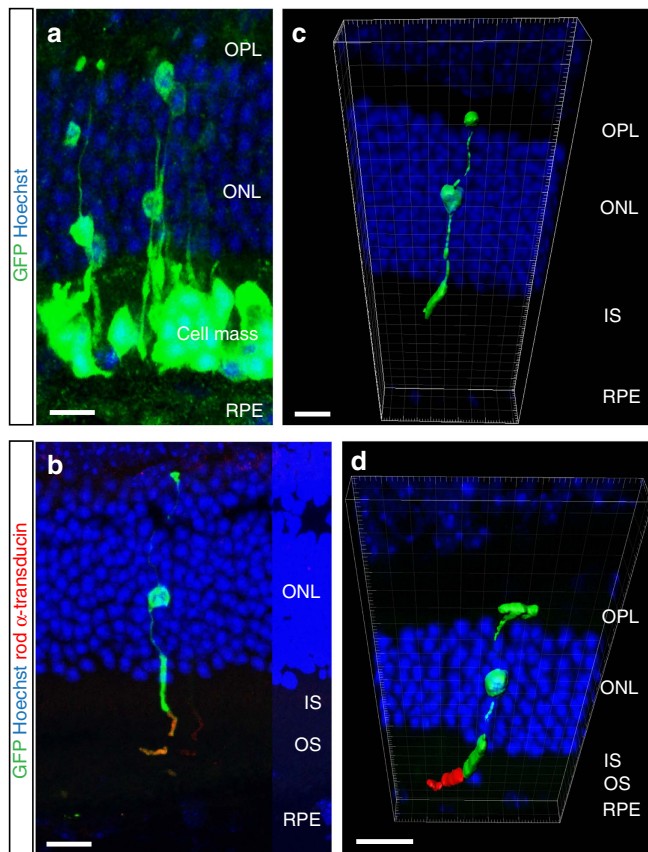

**Figure 8 | Material transfer does not require sustained physical interaction between donor and host cells.** (**a**), confocal image of GFP$^{+}$ cells within the host ONL immediately below the donor cell mass. Labelled user segments terminate close to the cell mass. (**b**), confocal image of GFP$^{+}$ cells within the host ONL a significant distance away from the cell mass, with no donor cells present in the overlying SRS. (**c,d**), 3D reconstructions of GFP$^{+}$ labelled cells in the ONL of (**c**) wild-type and (**d**) Gnat1$^{-/-}$ host retinae show no physical interaction with any other GFP$^{+}$ cell. Scale bars 10 µm.

we transplanted *Crx.GFP$^{+}$* ESC-derived photoreceptor precursor donor cells, which had previously been transduced with an ShH10.CMV.iCre gene therapy vector, into *tdTomato$^{floxed}$* reporter mice (Fig. 9). When cells expressing *tdTomato$^{floxed}$* acquire Cre recombinase, the LoxP sites are cleaved and the stop signal is excised allowing transcription of tdTomato. Because the signal is inducible, the incidence of false-positive signals should be low and fusion-type events would be seen as GFP$^{+}$/tdTomato$^{+}$.

Subretinal injection of ShH10.CMV.iCre virus into *tdTomato$^{floxed}$* mice confirmed the inducible expression of tdTomato (Fig. 9b). Conversely, no expression of tdTomato was observed in uninjected mice (Fig. 9c) or following subretinal injection of vehicle (Fig. 9d). To control for viral carry over from the donor cell preparation, the supernatant from the final cell wash was injected sub-retinally, as described previously[19,40]. Only a very small number of tdTomato$^{+}$ cells ($22 \pm 20$ photoreceptors per eye; $N = 4$), and no GFP$^{+}$ cells, were observed in each retina (Fig. 9e). This last observation underlines the point that material transfer is highly unlikely to be the result of uptake by the host cells of free nucleic acid released by donor cells damaged during the preparation process.

We next transplanted Cre-expressing *CrxGFP$^{+}$* donor cells; an average of $2,119 \pm 1,575$ GFP$^{+}$ cells ($N = 5$) were found within

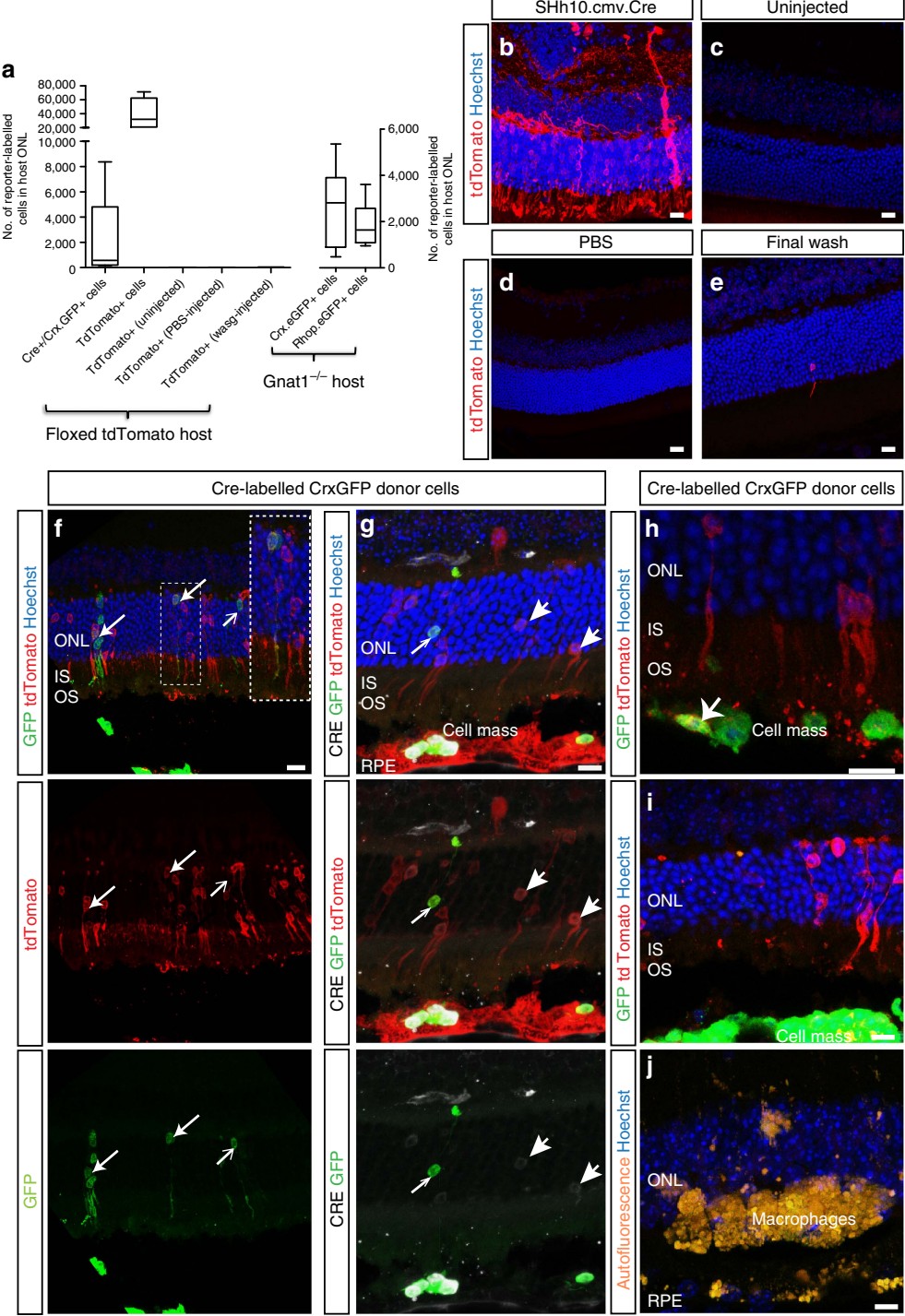

**Figure 9 | Transplantation of Cre + photoreceptor precursors leads to widespread expression of floxed reporters in host cells. (a)**, box (25–75% percentile) and whiskers ($_{min}$/max) plot showing median (line) number of reporter-labelled photoreceptors in adult *dTomatofloxed* host retinae following transplantation. **$P < 0.01$ 2-way ANOVA. N.B. all controls (uninjected, PBS-injected and final wash) were highly significantly different from both Cre$^+$/GFP$^+$ and TdTom$^+$ populations but are not denoted on graph for clarity). **(b–d)**, confocal projection images showing retinal sections after sub-retinal injection of **(b)** ShH10.CMV.iCre ($5 \times 10^{12}$ viral particles per ml$^{-1}$), **(c)** uninjected, **(d)** PBS-injected, or **(e)** mice injected with final wash from cell preparation. **(f)**, confocal projection images of GFP$^+$ (*green*) and tdTomato$^+$ (*red*) cells within the floxed tdTomato host retinae following transplantation of Crx.GFP$^+$/Cre$^+$ donor cells. Note that there are significantly higher numbers of tdTomato$^+$ cells than GFP$^+$ cells. *Inset*, example of a GFP$^+$, tdTomato$^+$ cells, surrounded by GFP$^-$/tdTomato$^+$ cells. **(g)**, Immunostaining for CRE (*grey*) shows robust protein expression in donor cells in SRS and some weaker expression in reporter-labelled cells within host retina. **(h)**, some cells in the SRS cell mass were tdTomato$^+$ (*arrow*), indicating the potential for bidirectional material transfer. For panels **(f)** and **(g)**, images from top to bottom show individual and/or combined channels for the same region of interest. **(i)**, tdTomato$^+$ cells were present in areas where donor cell masses were present but no GFP$^+$ cells within the underlying host ONL. **(j)**, few, if any, GFP$^+$ or tdTomato$^+$ cells were seen in retinae where donor cell mass had been rejected ($N = 2$ retinae). Scale bars 10 μm.

the recipient ONL, a level similar to that found using this donor line (without prior transduction with ShH10.CMV.iCre virus) into $Gnat1^{-/-}$ hosts (2,698 ± 591 GFP$^+$ cells; $N = 7$). Many of these cells also expressed tdTomato (Fig. 9f), indicating that they were host photoreceptors that received Cre Recombinase by material transfer. Note that Cre was introduced into donor cells using a virus and so not all GFP$^+$ donor cells would have been transduced. The presence of GFP$^+$/tdTomato$^-$ cells within the recipient ONL therefore reflect either true integration events or material transfer from a GFP$^+$/Cre$^-$ donor cell. In addition, we also observed small numbers of GFP$^+$/tdTomato$^+$ cells within the cell mass in the SRS (Fig. 9h). This provides further evidence that material transfer can also occur from host to donor, although this appears to be less frequent.

Strikingly, in addition to GFP$^+$/tdTomato$^+$ cells, we observed 18 times more photoreceptors that express tdTomato alone (GFP$^-$/tdTomato$^+$; 39,755 ± 9,888 cells) (Fig. 9a,f,g). TdTomato$^+$ cells were frequently seen in regions immediately below the donor cell mass, even in the absence of GFP$^+$ cells in the host retina (Fig. 9i). This is highly unlikely to be due to viral carry over: (i) injection of final wash led to little, if any, labelling of host cells (Fig. 9e), (ii) staining for CRE protein showed robust labelling of the majority of donor cells in the SRS and some, but not all, tdTomato$^+$ cells in the host ONL were CRE$^+$ but expression was weaker than in donor cells in the SRS (Fig. 9g) (all tdTomato$^+$ cells would be expected to be robustly CRE$^+$ if virally-transduced), (iii) transplantation of ESC-derived precursors labelled with the same capsid (ShH10.Rhop.eGFP), led to 1,745 ± 318.7 ($N = 9$) reporter-labelled cells in the host retina, very similar to that seen for Crx.GFP ESC-derived photoreceptors and more than a magnitude lower than the number of tdTomato$^+$ cells seen and (iv) tdTomato labelling was absent in eyes in which the cell mass had been rejected (Fig. 9j; $N = 2$).

That so many more cells in the host retina were GFP$^-$/tdTomato$^+$ than GFP$^+$/tdTomato$^+$ suggests that material transfer is a widespread phenomenon, but may not be sustained in all cells at levels sufficient to maintain expression of proteins such as GFP. Conversely, once sufficient levels of Cre Recombinase are present and Cre recombination occurs, the tdTomato reporter will be expressed permanently. It therefore reflects all previous recombination events, rather than the number of material transfer events occurring at that point in time.

## Discussion

Rescue of retinal degeneration by the transplantation of healthy photoreceptors has received considerable interest in recent years. Numerous studies have reported that transplantation of reporter-labelled donor photoreceptor cells, typically obtained from developmental stages equivalent to the post-mitotic precursor stage, into the adult wild-type and diseased retina leads to the presence of reporter-labelled cells in the host retina[9,12–14,19,27,37,41,42]. These apparently integrated photoreceptor cells were understood to arise from their migration into and integration within the host retina. Here, we present data to show that while donor cell migration and integration occurs, it reflects only a small proportion of events leading to the presence of donor-reporter-labelled cells within the host retina. Instead, we propose that post-mitotic donor and host photoreceptors engage in the transfer of cellular material, either as RNA and/or proteins, resulting in a wide variety of donor-derived proteins, including structural ones such as peripherin-2 and rhodopsin[9,11,13,19], being present in host photoreceptors.

This process of material transfer accounts for the majority of reporter-labelled cells within the host retina and raises the need to re-evaluate the cellular mechanisms underlying photoreceptor transplantation and their relative contributions to rescue of retinal degeneration. Almost all reporter-labelled cells in the host retina were additionally immunopositive for proteins otherwise expressed only by donor cells (for example, rod α-transducin in $Gnat1^{-/-}$ recipients[12,27], Peripherin-2 in $Prph2^{rd2/rd2}$ recipients[13,19]), which indicates that material transfer is a robust process that renders the majority of host cells undergoing material transfer functional, at least for a period of time. However, rather than integrating, the donor cells appear to be able to engage in transfer of a broad spectrum of gene products, through the transfer of RNA and/or proteins.

The cellular mechanisms underlying material transfer remain to be determined and will likely need to account for an extraordinary number of experimental findings, past and present. The mechanism does not involve classic nuclear fusion, as indicated by FISH, or the transfer of free protein or free nucleic acid. It does not require sustained physical interactions between donor and host photoreceptors, as demonstrated by the frequent presence of isolated GFP$^+$/rod α-transducin$^+$ cells in the host $Gnat1^{-/-}$ retina with no donor cells in the overlying SRS. Similarly, 3D reconstructions of such cells show only a single cell, with no evidence of cell–cell fusion or sustained physical contact. The Cre/LoxP experiments lend further evidence in support of this; the number of tdTomato$^+$ cells within the host retina far exceeds that of GFP$^+$ cells. Host cells bearing floxed tdTomato sites will require Cre Recombinase at a single moment in time to undergo recombination and permanently express tdTomato. Conversely, GFP's half-life[38] means that host photoreceptors would require a near-constant supply of GFP mRNA or protein to express it at levels sufficient to be detected by fluorescence imaging (or flow cytometry). Our data indicates a mechanism whereby host cells receive donor cell-derived material in a transient, but frequent, manner. The duration of rescue of any particular gene or protein is likely to be influenced by the long-term viability of the donor cell mass and true integrated cells as well as the half-life and turnover of the protein in question. The mechanism is a feature specific to photoreceptor precursor donor cells, rather than more immature cells, in accordance to previous reports[9,14,19], and does not appear to involve the release of either proteins or nucleic acid into the extracellular milieu. Finally, the mechanism must also explain a wide range of data that previously supported the notion of donor cell integration, including the ability to manipulate the number of donor-reporter-labelled cells within the host by altering structural aspects of the host retina such as OLM integrity[13,28,29] and chondroitin sulphate proteoglycan deposition[13,43,44] and by improving donor cell survival using the overexpression of growth factors[45].

The majority of reports describing fusion-type events have typically involved the fusion of a stem cell with a differentiated cell and the acquisition of the stem cell's nucleus by the host cell and formation of a bi-nucleated cell[20], a feature that was never observed here. Intriguingly, however, hematopoietic stem cells have been proposed to contribute genetic information in a transient manner to Purkinje cells. Nern et al.[46], proposed that nucleic acid and/or protein transfer between hematopoietic cells and Purkinje cells occurred either through transient fusion or intercellular vesicular transport mechanisms. Indeed, the vesicular transfer of DNA, mRNA and even organelles from bone marrow-derived cells to various different tissues has been reported recently[47] and may represent an important method of tissue repair. Most recently, exosomes, a type of microvesicle, have been described to mediate the transfer of mRNA, miRNA and protein from oligodendrocytes to neurons in the undamaged brain in an activity-dependent manner[48]. At the present time, it is

not possible to conclude whether material transfer between donor and host photoreceptors primarily involves the exchange of nucleic acid or protein, or even a combination of the two. While we can reasonably reject the notion that it involves the uptake of either free protein or free nucleic acid, it is tempting to speculate that microvesicles, or other structures that can similarly package nucleic acid and proteins, might play a role in material transfer between donor and host photoreceptors.

This study shows that material transfer must now be taken into account when designing applications for clinical translation. If the underlying cellular mechanisms of material transfer can be elucidated, they represent a novel therapeutic approach for introducing functional proteins into otherwise diseased photoreceptors, equivalent to a broad-spectrum gene-replacement approach. It will be particularly interesting to determine if human photoreceptor precursors can engage in material transfer. More immediately, however, cell-replacement therapy is most likely to be utilized in end-stage disease where there are very few photoreceptors remaining in the area of transplantation[13,49]. It will be particularly important to determine the extent to which visual function can be rescued by the transplantation of human photoreceptors into severely degenerated retina. In such situations, material transfer to host photoreceptors cannot take place and significant re-wiring of the host inner retina may have taken place[50].

## Methods

**Animals.** C57Bl/6 (Harlan, UK), Gnat1[−/−] (J. Lem, Tufts, Boston)[51], Prph2[rd2/rd2], (G. Travis, University College Los Angeles)[31], B6.Cg-Tg(Nrl-EGFP)1Asw/J (NrlGFP[+/+]) (A. Swaroop, University of Michigan)[30] and B6.Cg-Tg(CAG-DsRed* MST)1Nagy/J (DsRed[+/−]) and B6.Cg-Gt(ROSA)26Sor[tm14(CAG−tdTomato)Hze]/J (tdTomato[+/+]) (The Jackson Laboratory) were maintained on a standard 12 h light-dark cycle. Mice received food and water ad libitum and were provided with fresh bedding and nesting daily. NrlGFP and DsRed mice, as appropriate, were used at P8 (±1 day) for the provision of donor-derived post-mitotic photoreceptor precursors. All recipient animals were ∼6–12 weeks at the time of cell transplantation. Both male and female donor and recipient animals were used in all experiments, with the exception of FISH, where males and females were used as specified. All experiments have been performed in accordance with the United Kingdom Animals (Scientific Procedure) Act of 1986 and Policies on the Use of Animals and Humans in Neuroscience Research.

**Surgery and transplantation.** Mice were anaesthetized with an intraperitoneal injection of a mixture of Dormitor (1 mg ml[−1] medetomidine hydrochloride), ketamine (100 mg ml[−1]) and sterile water in the ratio 5:3:42. Pupils were dilated using 1% tropicamide and a topical anaesthetic was applied (Tetracaine). Eyes were protected with Viscotears (Novartis Pharmaceuticals, UK) and a glass coverslip placed over the eye. Surgery was performed under direct visual control using an operating microscope. A sterile 34-gauge hypodermic needle was used to make a small puncture to the anterior chamber to relieve pressure in the orbit. The same needle was used to slowly inject 1 μl of cell suspension into the sub-retinal space, between the neural retina and the RPE, in the superior ocular quadrant[12]. The needle was left in place for ∼20 s to allow for re-equilibration of intraocular pressure before slowly withdrawing. Anaesthesia was reversed using an equal amount of Antisedan (Pfizer Pharmaceuticals) and the eyes protected with Viscotears. Mice were placed on heat mats and received softened food until fully recovered.

**Preparation of donor cells from postnatal mice.** Neural retinae were isolated from P8 NrlGFP[+/+] or DsRed[+/−] mice or E11 eGFP[+/−] mice and a single-cell suspension was obtained using papain digestion, as reported previously[9,12]: Reagents were made up according to manufacturer's (Worthington Biochemical) instructions and comprised (i) papain solution (20 units of papain per ml, 1 mM L-cysteine and 0.5 mM EDTA, with DNase added at 100 units per ml), (ii) DNase solution (2,000 units per ml) and (iii) ovomucoid inhibitor (OMI) (10 mg OMI and 10 mg albumin per ml). Dissected neural retinas were incubated in papain solution at 37 °C, 95% O$_2$, 5% CO$_2$ for 45 min, with occasional gentle mixing. Samples were gently triturated and passed through a nylon cell strainer, yielding a single-cell suspension. The suspension was centrifuged for 5 min at 200g and the resultant pellet was re-suspended in EBSS with 10% OMI solution and 5% DNase solution to inhibit the papain reaction. After 5–10 min incubation, the suspension was layered on top of 500 μl of neat OMI solution and centrifuged for 5 min at 100 g to fully inhibit the dissociation reaction, and separate out debris and cell membranes.

After discarding the supernatant, the resultant cell pellet was re-suspended to a concentration of 10–20 million cells in EBSS containing fetal calf serum (FCS; 1%) and DNase solution (5%) ready for FACS.

Cells were FACS sorted for GFP or DsRed fluorescence using a BD Influx Cell Sorter (BD Biosciences) fitted with a 200 mW 488 nm blue laser to excite GFP. GFP[+] cells were identified using a 530/40 nm detector. A 70-micron nozzle at 50 p.s.i. was used and cells were collected on a 1:1 FBS/EBSS solution. Sorted cells were re-suspended at 200,000 live cells μl[−1] (as assessed using a Scepter hand-held cell counter; Millipore) in sterile EBSS and DNase solution (5%) and kept on ice before injection.

**FACS of donor cells from postnatal mice.** In experiments where a mix of GFP[+] and DsRed[+] donor cells was used, GFP[+] and DsRed[+] donor precursors were dissociated as above and isolated using the cell-surface marker, CD73 (refs 52,53). An antibody against mouse CD73 (APC-conjugated rat IgG1, clone TY/11.8, Miltenyi Biotec) was added at a 1:75 dilution and incubated for 30 min at 4 °C after dissociation and before FACS. Sorted CD73[+]/GFP[+] and CD73[+]/DsRed[+] cells were mixed post-FACS, before being transplanted.

**Preparation of donor cells from mouse ESCs.** A Crx.GFP mouse ES cell line (a kind gift of Professor Yvan Arsenijevic)[14] was maintained in GMEM containing 10% KSR (knockout serum replacement), 1% fetal bovine serum (FBS), 0.1 mM non-essential amino acids (NEAA), 1 mM pyruvate, 0.1 mM 2-mercaptoethanol with 2,000 U ml[−1] LIF, 0.5 μM MEK inhibitor (PD0325901) and 1.5 μM GSK3 inhibitor (CHR99021) (ref. 19). Briefly, for 3D retinal differentiation, $3 \times 10^4$ dissociated ES cells were re-suspended per millilitre of differentiation medium (GMEM containing 1.5% KSR, 0.1 mM NEAA, 1 mM pyruvate, 0.1 mM 2-mercaptoethanol), plated into 96-well low-binding plates and incubated at 37 °C, 5% CO$_2$. Embryoid body cell aggregates (EBs) formed within 24 h, on day 1 of culture, growth factor reduced Matrigel (GIBCO) was added to each well to give a final concentration of 2%. For whole-EB retinal differentiation toward photoreceptor cell fate, EBs were transferred into retinal maturation medium (DMEM/F12 Glutamax media containing N2 supplement and Pen/strep, herein Retinal Maintenance Media, RMM) at day 9, plated in low-binding plates at a density of 6 EBs cm[−2] and incubated at 37 °C, 5% CO$_2$. The media was changed every 2–3 days, with the addition of 1 mM Taurine and 500 nM retinoic acid from day 14 of culture onwards.

**Production of recombinant AAV and transduction of mouse ESCs.** Codon-improved Cre (iCre) recombinase-coding sequence was cloned into a pD10.CMV backbone. The resulting pD10/CMVpromoter-Cre construct, containing AAV-2 inverted terminal repeat, was used to generate ShH10.CMV.iCre virus. The expression cassette was packaged into recombinant AAV viral particles of ShH10 serotype[54] by a three-plasmid system, as described previously[55]. Viral particles (named ShH10.CMV.iCre) were purified through an AVB Sepharose column and concentrated to a final volume of 200 μl using Vivaspin 4 (10 kDa) concentrators. Viral particle titres were determined by quantitative PCR as described elsewhere[56] and expressed as viral genomes per ml. At day 22 of culture Crx.GFP EBs were infected with $1.2 \times 10^{11}$ viral particles per well in retinal differentiation medium. Estimated gMOI ∼4,000.

**FACS of mouse ESC-derived photoreceptors.** For transplantation, Crx.GFP/ shH10.iCre wEBs were dissociated at day 26–27 of culture into a single-cell suspension using a modified protocol using reagents from a papain-based Neurosphere Dissociation Kit (Miltenyi Biotec, 130-095-943). Briefly, samples were incubated in a papain-based enzyme dissociation mix (Miltenyi Biotech) at 37 °C for 15 min, gently triturated and then spun down at 320 g for 7 min at room temperature (RT). Cell pellets were re-suspended in HBSS (with 0.5 mM MgCl$_2$) with FCS (1%), 66% 25 mM HEPES and DNase solution (1%) and passed through a cell strainer ready for FACS.

Cells were FACS sorted for GFP fluorescence using a BD Influx Cell Sorter fitted with a 200 mW 488 nm blue laser to excite GFP. GFP was collected using a 530/40 nm detector. A 70-micron nozzle at 50 p.s.i. was used and cells were collected in a 20% FBS in RMM solution. Sorted GFP[+ve] cells were re-suspended at 200,000 live cells μl[−1] in sterile HBSS (+Ca[2+], Mg[2+]) and DNase solution (3%) and kept on ice before injection. The supernatant from the final spin before re-suspension was retained for control injections (see the 'Results' section). After injection of supernatant, in addition to a small number of tdTomato[+] photoreceptors, tdTomato[+] RPE (26 ± 21 cells), Muller Glia (1 ± 2 cells) and other inner retinal cells (2 ± 3 cells) were observed.

**Preparation of GFP[+] donor fibroblasts.** An NIH3T3/GFP cell line (AKR-214, Cell Biolabs) was grown in DMEM (high glucose) with 10% FBS, 0.1 mM MEM NEAA, 2 mM L-glutamine and 1% Pen-Strep in a 37 °C incubator at 5% CO$_2$. Cells were passaged every 3–4 days. Before transplantation, cells were dissociated using 0.05% trypsin, washed with phosphate-buffered saline (PBS) and re-suspended at 50 000 cells μl[−1] in sterile EBSS and DNase solution (5%) and kept on ice before

injection. A volume of 1 µl of the cell suspension was injected sub-retinally, as described above.

**Preparation of rEGFP.** rEGFP purified from *E. coli* was reconstituted in sterile PBS at 1 mg ml$^{-1}$. A volume of 1 µl was injected into the SRS, as described above.

**Cell counts.** Eyes were collected 5–6 weeks post transplantation, unless otherwise stated, and cryoembedded before sectioning and mounting. GFP$^+$ and/or DsRed$^+$ cells were located using epifluorescence illumination. The average number of reporter-labelled cells per eye was determined by counting every third section and multiplying by three to give a total/eye, as described previously[12]. The majority of experiments did not require blinded assessment since no test/control comparisons were being made. Where relevant, however, counts were made in a blinded manner, whereby the assessor was unaware of the donor cell source for any given eye. Cell counts for individual eyes were excluded from the analysis if there were cells in the vitreous, indicative of accidental intravitreal transplantation of the cells, if there was no cell mass present in the SRS, indicative of reflux at time of injection, and/or there was significant macrophage infiltration and evidence of level II/III rejection, as defined in ref. 37.

**Histology and immunohistochemistry.** Eyes were fixed in 4% paraformaldehyde for 1 h before cryoprotecting in 20% sucrose and embedding in OCT (RA Lamb) and freezing in isopentane precooled in liquid nitrogen. Cryosections were cut at 18-µm thick and all sections were collected for analysis. For histology and cell counting, cryosections were air-dried for 30 min and washed in PBS containing Hoechst 33342 (10 µM) for 5 min before mounting. For immunohistochemistry, sections were blocked in 5% goat serum and 1% bovine serum albumin and triton X-100 (0.05% in PBS at RT for 1 h. Primary antibody was incubated overnight at 4 °C. After washing, sections were incubated with secondary antibody for 2 h at RT, washed and counter-stained with Hoechst 33342 (10 µM). Primary antibodies used included: LaminB antibody (mouse monoclonal; Abcam Ab8980; 1:200); rod α-transducin (rabbit polyclonal; Santa Cruz SC389; 1:1,000); CRE Recombinase (Mouse monoclonal; Millipore MAB3120; 1:200). Alexa fluor 488, 546 and 633 secondary antibodies (Invitrogen-Molecular Probes) were used at a 1:500 dilution. Negative controls omitted the primary antibody. Retinae receiving rEGFP were examined both unstained, as is typical for assessing retinae receiving retinal transplants[9,12] and following immunostaining with a directly conjugated chicken anti-GFP antibody (Invitrogen-Molecular Probes; 1:200 for 2 h before mounting).

**Fluorescent *in situ* hybridization.** Eye-cups were fixed for 1 h in 4% paraformaldehyde and embedded in paraffin. Sections were cut (4 µm thick) and screened for GFP-positive cells. For FISH, sections were dewaxed in Histoclear for 10 min and brought to water. Pressure cooker permeabilization was performed for 2 min in unmasking solution, low pH (Vector Labs). Sections were treated with 0.05 mg ml$^{-1}$ of Protease K solution for 20 min, at 37 °C. After washed and dehydrated, slides were left to air dry. In all, 10 µl of Y-chromosome paint probe (Empire Genomics) was added to each 22 × 22 mm area and sealed. Following denaturation, for 5 min at 95 °C, sections were incubated overnight at 37 °C. After seal removal slides were placed in 2 × SSC for 10 min at RT and then incubated with 0.5 × SSC for 2 min at 73 °C. Samples were blocked in 3% goat serum, 1% bovine serum albumin, 0.01% Triton in PBS for 30 min at RT. Primary antibody (polyclonal chicken anti-GFP; Abcam) was incubated for 6 h at RT. Sections were incubated with secondary antibody for 2 h at RT, washed and counter-stained with Hoechst 33342 (10 µM).

**Confocal microscopy.** Retinal sections were viewed on a confocal microscope (Leica TCS SPE, Leica Microsystems). Unless otherwise stated, retinal images show merged projection images of an xyz confocal stack through retinal sections, ~18 µm thick, as stated. Individual xy images were acquired using a 2-frame average and at 1,024 × 1,024 resolution and at ~1 µm step intervals throughout the depth of the stack.

For assessment of co-localization, the pinhole was set to the minimum possible to permit consistent DsRed signal but minimize the potential for contaminating fluorescence signal from above or below the imaged plane. The same laser intensity, gain and offset settings were used to image all samples. GFP$^+$ cells located within the recipient ONL were selected using the GFP channel alone, without prior knowledge of signal in the red channel, and on the basis of clear morphology, including cell body and inner/outer segments. Two independent assessors then examined both the cell bodies and the inner segments, which have a comparatively large cytoplasmic volume, for increases in GFP fluorescence signal and concomitant changes in DsRed fluorescence signal. Measurements were made using the Leica analysis package and using the line tool to draw an ROI line. Line measurements were taken from 3 single confocal sections at different depths through the cell, giving 2 × 3 measurements per cell. A consensus outcome was assigned if 4/6 of the assessments were in agreement. In all, 3/6 or lower was taken as no consensus (<4% of readings, see the 'Results' sections). Please note, despite careful analysis, it was not always possible to be confident that a given inner segment was connected to a specific cell body. For this reason, the data should be regarded as assessments of two different regions of a cell, but not necessarily arising from the same cell.

**Isolation and analysis of transplanted eyes by flow cytometry.** Transplanted retinae were taken 5–6 weeks post transplantation. Neural retinae were isolated by dissection and any overlying cell mass carefully removed using direct visualization under a fluorescence microscope. Individual neural retinae were dissociated using the papain-based Neurosphere Dissociation Kit, described above. An aliquot from each sample was taken to determine absolute cell counts using the Beckman Coulter Vi-Cell XR Cell Viability Analyser (Beckman Coulter). The remaining media was spun down at 320 g for 7 min at 4 °C and the samples were subsequently stained in 100 µl of DMEM$^+$ media with or without an anti-mouse CD45-APC-Fire750 antibody (final concentration 2 µg ml$^{-1}$) (BioLegend) for 30 min, in the dark, on ice. The samples were then spun down at 320 g for 7 min at 4 °C, aspirated and then re-suspended in 350 µl of DMEM$^+$ media with 1.5 µM of Sytox Blue dead cell stain for sample acquisition on the flow cytometer.

All of the samples were acquired using a BD LSRFortessa X-20 flow cytometer (BD Biosciences), equipped with 5 lasers (that is, 355 nm, 405 nm, 488 nm, 561 nm and 640 nm lasers). The entire sample was acquired for analysis due to the low total frequency of transplanted donor cells. Fluorescence gates were set using NrlGFP and DsRed, and wild-type retina as positive, and negative, controls, respectively (see Fig. 2b–d and Supplementary Fig. 1).

**Real-time imaging.** Rod precursor cells were transplanted into 12-week old *Prph2$^{rd2/rd2}$* mice as described above. *Prph2$^{rd2/rd2}$* mice were used as hosts, since they still retain a moderately robust ONL but lack outer segments, which otherwise cause significant attenuation of the fluorescence signal from the underlying ONL (N. Aghaizu, unpublished data). At 3 days post transplantation, the eyes were collected into ice-cold Phenol-red free RPMI 1640 medium (ThermoFisher Scientific). Retinae were isolated free of surrounding tissues and exposed to 5 µM Mitotracker Orange CMTMRos (ThermoFisher Scientific) diluted in RPMI 1640 medium for 15 min at RT. Retinal whole mounts were prepared under a dissection microscope by flattening the retinae with the photoreceptor side up onto a 0.45 µm MF-Millipore nitrocellulose membrane. For time-lapse recordings, mounted retinae were placed in DMEM$^{gfp}$-2 live-imaging medium (Evrogen) supplemented with 5% FCS contained within a 60-mm Petri dish. To prevent specimen drifting, membranes were gently pressed onto a thin layer of adhesive vacuum grease and a platinum ring was additionally placed over the membrane. Images were acquired on a Leica SP8 upright confocal laser scanning microscope fitted with a 25 × water-immersion objective (NA = 0.95) as well as Leica photomultiplier tube/avalanche photo diode hybrid HyD detectors. The Chameleon Compact OPO multiphoton laser source (Coherent) was tuned to a wavelength of 900 nm for the excitation of both GFP and Mitotracker Orange. The pinhole was fully dilated. All recordings were made in a temperature and gas controlled environmental chamber set to 37 °C and 5% CO$_2$. For image acquisition, *xyzt* time-lapse image stack series were captured at a resolution of 512 × 512, at a step size of 1 µm and at 20 min intervals.

**Statistical analysis.** All means are presented ± s.d., unless otherwise stated; N, number of retinae; n, number of cells. All transplantation cell count data is based on at least two independent transplantation runs. Each run therefore involves independent litters of donor animals, cell preparation and FACS sorting, surgery and different litters of host animals. D'Agostino and Pearson's test was used to determine normality of data sets, and the appropriate statistical tests applied, as required. The statistical program used for analysis was GraphPad Instat 3 (GraphPad Software, Inc, La Jolla, USA), *P < 0.05, **P < 0.01.

**Data availability.** The authors declare that the data supporting the findings of this study are available within the article and its Supplementary Information files and from the corresponding author on reasonable request.

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

## Acknowledgements

This work was supported by the Medical Research Council UK (MR/J004553/1), European Research Council (ERC-2012-ADG_20120314), Fight For Sight (1448/1449), the Macular Vision Research Foundation, The Miller's Trust, the Special Trustees of Moorfields Eye Charity and a generous donation by Mr Otto van der Wyck. R.A.P. is a Royal Society University Research Fellow. A.G.-C. is UCL Sensory Systems and Therapies Fellow. N.A. is a UCL Grand Challenge Ph.D student. D.G. is a Singapore A Star PhD student. K.K. is a Wellcome Trust 4 year Stem Cells and Development PhD student. P.V.W. is an MRC-funded PhD student on the Wellcome Trust 4 year Neuroscience PhD programme. J.C.S. is supported by Great Ormond Street Hospital Children's Charity and the National Institute for Health Research Biomedical Research Centre at Great Ormond Street Hospital. R.R.A is partly funded by the Department of Health's National Institute for Health Research Biomedical Research Centre at Moorfields Eye Hospital. R.R.A and R.A.P are in part funded by Alcon Research Institute. We thank S. Azam, R. Maswood and A. Michacz for virus purification, L. Abelleira Hervas for assistance with animal breeding, M. Carandini for the donation of mice, and M. Rizzi and other members of the department of genetics for constructive discussions.

## Author contributions

R.A.P. contributed to the conception, design, execution and analysis of all experiments, funding and wrote the first draft of the manuscript. A.G-C. contributed to the conception, design, execution and analysis of all experiments, and writing of the manuscript. E.L.W contributed to the conception, design, execution and analysis of several experiments and manuscript preparation. J.R.C-R. performed the Fluorescence *In Situ* Hybridization and contributed to histological processing. N.A. designed and carried out the real-time imaging experiments and contributed to manuscript preparation. D.G., P.V.W. and K.K. contributed to the execution of a number of the experiments. R.D.S. performed FACS sorting and analysis. A.G. contributed to

experimental design and viral constructs, and contributed to manuscript preparation. Y.D. contributed to histological processing and surgery. A.N. and M.K. contributed to the maintenance of stem cell cultures and viral production. E.C. contributed to viral construct design and production. K.W-C. contributed a data set for secondary analysis. J.C.S. contributed to the interpretation of experiments, manuscript writing and funding. A.J.S. contributed to the conception, design and interpretation of experiments, and manuscript writing. R.R.A. contributed to the conception, design and interpretation of experiments, manuscript writing and funding.

## Additional information

**Competing financial interests:** The authors declare no competing financial interests.

