## [Peer Review File · Nature Communications]

Reviewer #1 (Remarks to the Author)

The paper by Pearson et al. titled "Donor and host photoreceptors engage in material transfer following transplantation of post-mitotic photoreceptor precursors" found a key problem with the interpretation of previous photoreceptor cell transplantation experiments to retina that were originally reported by the same group. In their previous experiments transplanted cells were labeled with a marker protein, and retinal integration was assessed by following the location and connectivity of cells marked with the marker protein. In a convincing set of experiments Pearson and colleagues show that marker proteins can move from donor cells to host cells, making previous conclusions about integration of photoreceptors to the retinal circuit incorrect. They also show that it is the protein and not mRNA that moves between the donor and host cells. This is a very important paper that will cause a major reevaluation of results in the field of cell transplantation.

The paper is performed in high standard and I do not see any problem with the data and the interpretation of the data. I recommend publishing this paper immediately without any modifications. The authors should be congratulated to be honest to submit this work that partly contradict their previous paper, but clears the field from future misinterpretations.

Reviewer #2 (Remarks to the Author)

In their manuscript titled "Donor and host photoreceptors engage in material transfer following transplantation of post-mitotic photoreceptor precursors" Pearson and colleagues describe the cellular mechanisms of photoreceptor integration into the retina. They report that there may be exchange of cellular components between the donor and host cells using GFP in the donor and RFP in the host. They used both imaging and flow sorting to analyze the co-localization of GFP and RFP. To determine if genetic transfer could occur, they performed FISH for Y chromosome and use male donor cells and female host. They found that only a small number of integrated photoreceptors has the Y chromosome suggesting that there was transfer of the GFP transgene or protein from the donor to the recipient rather than functional integration for the majority of cells. Indeed, they went on to show that other proteins could be transferred to host photoreceptors. They conclude that the protein is transferred and this is specific to photoreceptor precursors. Finally, they perform experiments with Cre recombinase donor cells and Cre-reporter host cells and show transfer of Cre activity.

Overall the experiments are well designed and executed and the data are convincing. My only comment is that they seem to favor the idea that proteins are transferred rather than nucleic acid. However, in the preparation of the cells, they use a papain digestion and massive amounts of donor DNA is released as part of that enzymatic processing. Without careful DNaseI treatment and subsequent washing of the cells, there will be high concentrations genomic DNA fragments in the donor cell prep. In their materials and methods, they do not mention DNaseI treatment as is done in other cellular preparations. While they formally consider the possibility of DNA as the key element, they did not perform experiments to address this. They need to analyze the amount of DNA in their cellular prep, treat with DNaseI and carefully wash the cells and then look for transfer of DNA to the host cells. I believe it is much more likely to be transfer of DNA than protein given the half life of the proteins being used in this study.

Reviewer 1

The paper by Pearson et al. titled "Donor and host photoreceptors engage in material transfer following transplantation of post-mitotic photoreceptor precursors" found a key problem with the interpretation of previous photoreceptor cell transplantation experiments to retina that were originally reported by the same group. In their previous experiments transplanted cells were labeled with a marker protein, and retinal integration was assessed by following the location and connectivity of cells marked with the marker protein. In a convincing set of experiments Pearson and colleagues show that marker proteins can move from donor cells to host cells, making previous conclusions about integration of photoreceptors to the retinal circuit incorrect. They also show that it is the protein and not mRNA that moves between the donor and host cells. This is a very important paper that will cause a major reevaluation of results in the field of cell transplantation.

Point 1: *The paper is performed in high standard and I do not see any problem with the data and the interpretation of the data. I recommend publishing this paper immediately without any modifications. The authors should be congratulated to be honest to submit this work that partly contradict their previous paper, but clears the field from future misinterpretations.*

Response to point 1: Reviewer 1 had no concerns. See "Further Response" below regarding interpretation below.

Reviewer 2

In their manuscript titled "Donor and host photoreceptors engage in material transfer following transplantation of post-mitotic photoreceptor precursors" Pearson and colleagues describe the cellular mechanisms of photoreceptor integration into the retina. They report that there may be exchange of cellular components between the donor and host cells using GFP in the donor and RFP in the host. They used both imaging and flow sorting to analyze the co-localization of GFP and RFP. To determine if genetic transfer could occur, they performed FISH for Y chromosome and use male donor cells and female host. They found that only a small number of integrated photoreceptors has the Y chromosome suggesting that there was transfer of the GFP transgene or protein from the donor to the recipient rather than functional integration for the majority of cells. Indeed, they went on to show that other proteins could be transferred to host photoreceptors. They conclude that the protein is transferred and this is specific to photoreceptor precursors. Finally, they perform experiments with Cre recombinase donor cells and Cre-reporter host cells and show transfer of Cre activity.

Point 1: *Overall the experiments are well designed and executed and the data are convincing. My only comment is that they seem to favor the idea that proteins are transferred rather than nucleic acid. However, in the preparation of the cells, they use a papain digestion and massive amounts of donor DNA is released as part of that enzymatic processing. Without careful DNaseI treatment and subsequent washing of the cells, there will be high concentrations genomic DNA fragments in the donor cell prep. In their materials and methods, they do not mention DNaseI treatment as is done in*

other cellular preparations.

Response to Point 1: Free nucleic acids are unlikely to survive the FACS sorting and multiple washes involved in cell preparation, and the uptake of free DNA by eukaryotic cells is very poor. More importantly, we can confirm that DNaseI was included throughout the cell preparation process, including in the final injection buffer. We recognize that we were not sufficiently clear in the original submission in describing all the steps in the dissociation process, particularly those including DNaseI and washing steps, instead referring to previous publications. We have now explained the protocol in full in the Methods section.

Point 2: *While they formally consider the possibility of DNA as the key element, they did not perform experiments to address this. They need to analyze the amount of DNA in their cellular prep, treat with DNaseI and carefully wash the cells and then look for transfer of DNA to the host cells. I believe it is much more likely to be transfer of DNA than protein given the half-life of the proteins being used in this study.*

Response to Point 2: In addition to the presence of DNaseI throughout, as detailed above, we have already excluded the possibility that extracellular DNA present in the injection buffer (of resuspended cells) could lead to material transfer. In the Cre/Lox experiments described in Fig. 9, we directly injected the final supernatant of the cell preparation process. While this was primarily to assess viral carry over, it is the same experiment suggested by the reviewer and serves to test for any potential carry over of free nucleic acids in the extracellular buffer. In these experiments, we saw no GFP+ cells within the recipient ONL (see Fig. 9e). If carry over of released extracellular DNA were important for material transfer, this would have resulted in the presence of GFP reporter labeled cells in the host retina. This is now stated explicitly.

Further Response: The referee's comments do raise an important point regarding interpretation, which is that both referees suggest that we propose it is protein and not nucleic acid that transfers across. We would like to stress that we do not know at this point whether it is nucleic acid or protein (or, indeed, both) that travels across. That both referees came away with the same interpretation, however, means that we have not made this point clearly enough. To clarify, we formally excluded the possibility of wholesale transfer of DNA via nuclear fusion (by FISH). We have also demonstrated that it is highly unlikely that material transfer involves uptake of *free* protein, as shown by the failure to label cells with rEGFP, or *free* nucleic acid, for the reasons stated above. It remains possible that either protein and/or RNA could be transferred if packaged in vesicles or via transient cell-cell contacts, for example, and we have added further points to the Results and Discussion (highlighted in yellow) to emphasize our current understanding. In our original submission we had stated the transfer of "*RNA and/or protein*" throughout the manuscript, to underline this point (highlighted in blue) and we have now re-emphasized this throughout. As noted by the editors, at this time we can only speculate as to the cellular mechanism underlying such transfer of RNA and/or protein between photoreceptors, although we are working hard to uncover it.